# Bisulphite sequencing in the presence of cytosine-conversion errors

**Thomas James Ellis**[ID]**, Viktoria Nizhynska, Rahul Pisupati, Almudena Mollá-Morales, Magnus Nordborg**[ID]*

Gregor Mendel Institute, Austrian Academy of Sciences, Vienna BioCenter, Vienna, Austria

* magnus.nordborg@gmi.oeaw.ac.at

## Abstract

'Tagmentation' approaches to bisulphite sequencing use a transposase to simultaneously make double-stranded breaks and ligate adaptors to the resulting fragments, allowing for higher throughput with less starting material. However, it has also been noted that certain tagmentation protocols have an unusually high number unmethylated cytosines that are not converted to thymine. Here we describe this phenomenon in detail, and find that results are consistent with single strand nicks by the transposase, followed by strand displacement of part or all of the DNA fragment, leading to erroneous incorporation of methylated cytosines. Nevertheless we show that these errors can be accounted for in downstream analysis and need not impede biological conclusions. We provide a Python package to allow users to implement this framework. Ultimately the additional effort of accounting for errors must be traded off against the scalability of the protocol in planning experiments.

## Introduction

Cytosine methylation is a common epigenetic mark associated with heterochromatin and transcriptional silencing. A common approach to assay methylation is to treat DNA with sodium bisulphite to convert unmethylated cytosines to uracil that are in turn converted to thymine in a subsequent PCR step [1]. Methylated cytosines remain intact. Following sequencing of the resulting DNA fragments, reads are aligned to a reference genome. The methylation status at each site is determined based on how many thymines and cytosines align to each genomic cytosine position. This gives a quantitative estimate of the proportion of cells in which each cytosine is methylated.

A major limitation of bisulphite treatment is that the process itself damages DNA, meaning that substantial amounts of starting material are required for adequate results. Early studies provided the first genome-wide surveys of methylation profiles, but relied on sonication of DNA followed by a delicate adaptor-ligation step (e.g. [2–4]). Aiming to make the procedure more easily scalable to high-throughput samples, 'tagmentation' approaches use a Tn5 transposase to simultaneously cut the DNA and ligate adaptors and PCR primers to the fragments [5]. In the original protocol, the transposase is loaded with a single adaptor, and the second adaptor is annealed by by oligo replacement, followed by repair of the nine-base-pair single-stranded gap left by the transposase [6]. This allows for several orders of magnitude

**Data availability statement:** Sequencing data generated in this study are available in the National Center for Biotechnology Information (NCBI) Sequence Read Archive (SRA) under the

accession number PRJNA1155267. Code to run
the analyses is available from Github
(https://github.com/ellisztamas/bs_seq_with_
typing_errors) and Zenodo. Citations to the data
and code including DOIs are given in the
manuscript.

**Funding:** This work was supported by an ERC
Advanced Grant (789037) to Magnus Nordborg.
The funders had no role in study design, data
collection and analysis, decision to publish, or
preparation of the manuscript.

**Competing interests:** The authors have
declared that no competing interests exist.

less starting material than sonication-based protocols, but the oligo-replacement step remains
the most challenging step. Lu et al. [7] modified this approach to load the transposome with
two methylated adaptors, and to replace the oligo-replacement and gap repair steps with a
single strand displacement reaction by a *Bst* polymerase. The strand displacement step simul-
taneously fills the nine-base-pair gap and replaces the 3'-adaptor with a methylated oligonu-
cleotide containing the reverse sequencing primer (see also [8,9]). In contrast to the previ-
ous approach, gap repair is carried out using dNTPs with 5-methyl-dCTPs instead of dCTPs.
Since these gap regions are removed as part of the adaptors during bioinformatic process-
ing, these methyl-dCTPs should not in principle affect methylation of genomic DNA. Using
two adaptors also has the advantage that the original and complementary strands generated
by PCR can be distinguished by read-alignment software, effectively doubling coverage. The
stand-displacement tagmentation protocol thus has the potential to yield more information
from less starting material.

However, accurate inference of methylation states requires that unmethylated cytosines
be reliably converted to thymines. A standard quality-control step is to include unmethy-
lated control DNA, and to estimate the conversion rate as the proportion of cytosines that are
sequenced as thymines. In the ideal case, all unmethylated cytosines should be converted to
thymine. Lu et al. [7] and Suzuki et al. [9] reported that 1-2% of reads generated by a strand-
displacement protocol appeared to be entirely unconverted. Lu et al. [7] hypothesised that
this phenomenon is due to rare single-strand nicks in the 5' adaptors that serve as a start site
for extension by the *Bst* polymerase, leading to strand displacement over the whole fragment.
Because the protocol uses 5-methyl-dCTPs, this would cause the whole strand to become
methylated, potentially biasing or obscuring real methylation patterns. Lu et al. [7] and Suzuki
et al. [9] filtered out suspicious reads, but do not discuss the issue further.

Here we address this issue by providing a detailed description of non-conversion errors
using strand-displacement-based tagmentation. We first describe patterns of non-conversion
within reads and across the genome and a possible mechanism for their source. We describe a
statistical framework to account for non-conversion errors to quantify and classify real cyto-
sine methylation, and demonstrate that methylation levels can be inferred in the presence of
non-conversion errors.

## Results and discussion

### Non-conversion affects all or part of a read

We investigated how and where cytosine non-conversion errors occur in *Arabidopsis thaliana*,
*Drosophila melanogaster* and phage λ. Plant chloroplasts, and the genomes of fruit flies and
this virus lack cytosine methylation, so any unconverted cytosines we observe must be erro-
neous. We only examined the first half of the *A. thaliana* chloroplast, because the second
half shows strong homology with the autosomes. We used a modified version of the strand-
displacement protocol described by Weichenhan et al. [8] but varied the concentration of
transposase and number of PCR cycles to identify possible artefacts arising from these aspects
of the protocol. Overall cytosine non-conversion rates were very high, ranging from a few
percent up to more than 17% (Fig 1A). For comparison, methylation on chloroplasts in
*A. thaliana* is typically a fraction of a percent [10]. There was substantial variation between
libraries, but there was no consistent effect of species, transposase concentration, or num-
ber of PCR cycles. Since we used the same DNA preparation for each library, this is techni-
cal variation between libraries rather than biological variation. In particular, we note that the
highest non-conversion rates for phage λ were prepared as part of the same libraries as the
samples of *D. melanogaster* with the highest non-conversion rates, indicating that the effect

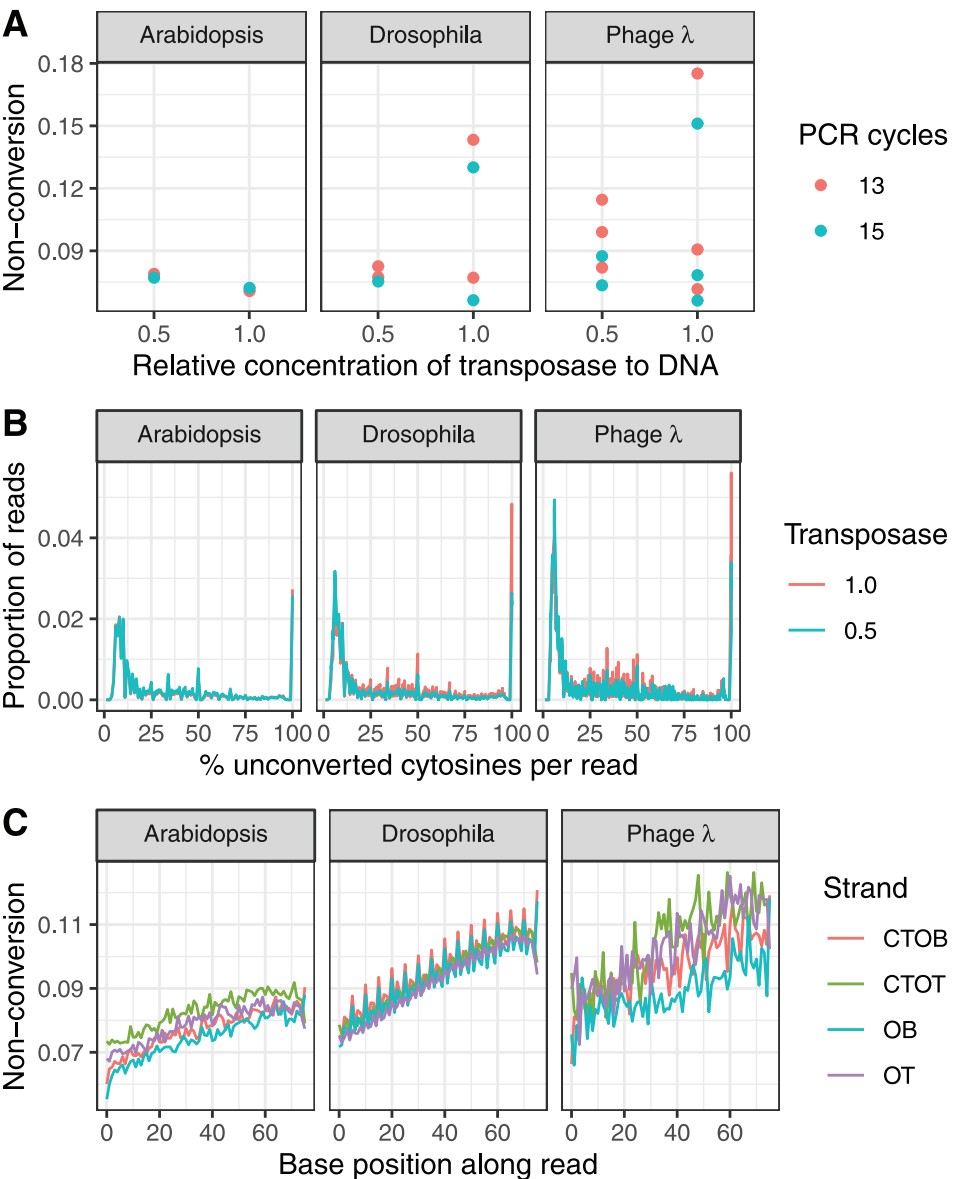

**Fig 1. Cytosine non-conversion within and between reads on unmethylated DNA from *A. thaliana*, *D. melanogaster* and phage λ.** For *A. thaliana*, data are shown for reads aligning to the first 75 kb of chloroplast. (**A**) Overall cytosine non-conversion rates in libraries prepared using different relative concentrations of Tn5 transposase to DNA, and 13 or 15 PCR cycles. (**B**) The distribution of the proportion of unconverted cytosines across reads. For clarity, reads with no unconverted cytosines are not shown. (**C**) The probability of observing an unconverted cytosine at each base0pair position within reads each of the four strands generated by bisulphite libraries (OT=original top strand; OB=original bottom; CTOT, complementary to original top; CTOB, complementary to original bottom) averaged over libraries.

is due primarily to differences between libraries. High cytosine non-conversion rates do not seem to be constrained to any species, and there is substantial stochasticity between samples.

To investigate the phenomenon further we examined the distribution of cytosine non-conversion within and between sequencing reads. Reads containing at least one cytosine fell into three categories. The majority of reads (71%, on average) showed no unconverted

cytosines at all. Remaining reads were roughly evenly split between those where every cytosine on a read was unconverted, and those where only a proportion of cytosines were unconverted (Fig 1B). The bimodal distribution of reads indicates that two distinct processes contribute to errant cytosine non-conversion. These observations contrast with previous reports of errant non-conversion in mammals which found that only 1-2% of reads being affected, and that when errors did occur, they occurred across the entire read [7,9].

We found that non-conversion errors in partly-methylated reads were more likely to occur towards the 3' than the 5' end of the read (Fig 1C). This may be partly due to the general tendency of sequence quality to decline along a read. However, this is unlikely to be the main cause in this case because Phred scores are consistently high across reads. It may also be that the presence of fully, and densely methylated cytosines in the adaptor sequences interfere with cytosine conversion. This is also unlikely, because in this case we would expect increased non-conversion at both ends of the read. This pattern must be due to reads with only partial non-conversion, since fully unconverted reads would increase non-conversion uniformly across the whole read.

There is also some evidence that non-conversion errors depend on strand. Bisulphite libraries typically distinguish between four strands: the original top and bottom strands (OT, OB), and the strands complementary to the original (CTOT) and bottom (CTOB) strands that are synthesised during PCR. In the *A. thaliana* and phage $\lambda$ samples there is a slight tendency for the top strands to have increased errors compared to the bottom strands, and for the complementary strands to have increased errors compared to the corresponding original strands. In the *D. melanogaster* samples there also appears to be a periodicity in the probabilities of non-conversion every five bases on the OB and CTOB strands only. The reasons for this apparent strand-specificity are unclear.

## Non-conversion errors vary across the genome

Taking the average cytosine non-conversion rate on control DNA gives a point estimate of the genome-wide non-conversion rate, but this might mask variation in error rates between regions of the genome. To investigate this we estimated non-conversion rates in 150 bp windows across the *A. thaliana* chloroplast, corresponding to the approximate size of one loop of a nucleosome. We found as much as five-fold variation in cytosine non-conversion between windows with remarkable consistency between samples (Fig 2A). One explanation for this is that there is much less data per window than for the whole chloroplast, so estimates at any individual window will have larger standard errors, inflating the variance between windows. However, the observed variance between windows was 6.4-fold greater than between simulated draws from a binomial distribution using the observed coverage at each window as the number of trial and the global mean non-conversion rate as the mean. Moreover, Spearman correlation coefficients ($\rho$) between replicates ranged from 0.71 to 0.84, showing that the pattern was remarkably consistent between technical replicates of the library preparation protocol (Fig 2A). We repeated this analysis for windows of 11 bp, corresponding to one helix turn around a histone, and found similar results. These observations indicate that the variation in non-conversion across the chloroplast reflects something intrinsic about the DNA rather than stochastic variation.

What might explain this variation? There was a weak negative correlation ($\rho$ between −0.29 and −0.10) between non-conversion and coverage at each window in three samples, with a weak positive correlation in the remaining sample ($\rho = 0.18$; Fig 2C). Nevertheless, while very low coverage causes estimates to be noisy, the lowest coverage per cytosine in any window was

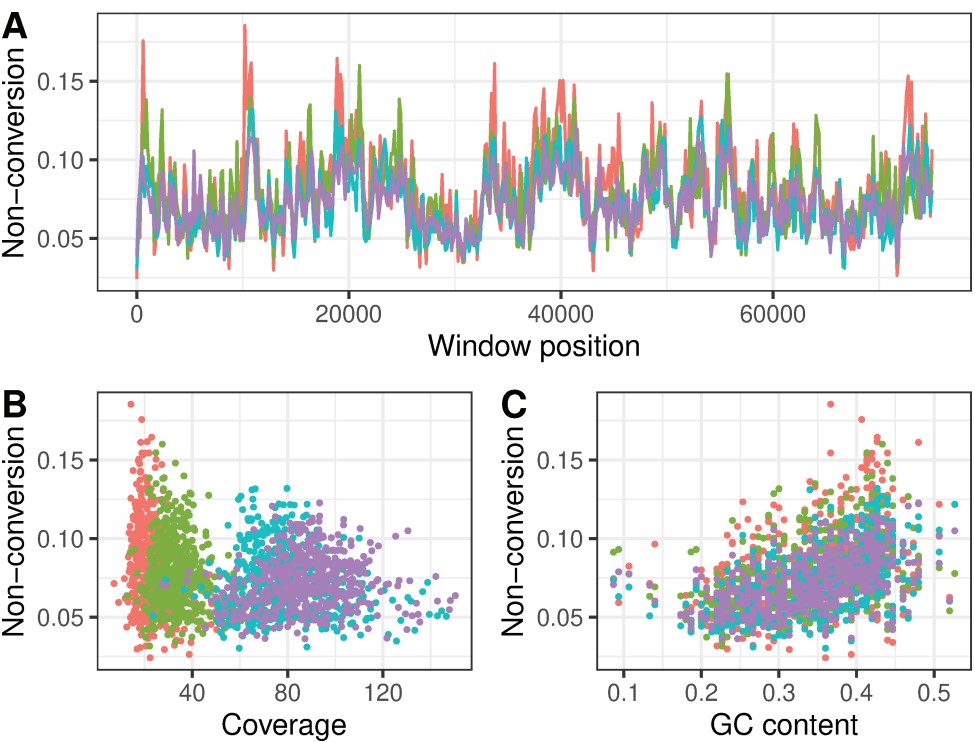

**Fig 2. Non-conversion errors vary at the kilobase scale.** Cytosine non-conversion across the first half of the chloroplast in four independent replicate library preparations of the *A. thaliana* accession Col-0. Colours denote independent libraries. (**A**) 150 bp windows across the chloroplast. (**B**) Relationship between non-conversion and coverage per base. (**C**) Relationship between non-conversion and GC content.

32.7×, so this cannot account for a large proportion of the variance. There were also consistent positive correlations between non-conversion and GC-content in each window across samples ($\rho$ between 0.43 and 0.55; Fig 2C). The density of CG, CHG and CHH sites explained 1.7%, 9.9% and 10.6% of the variation in (logit) non-conversion rate respectively. Since CG are the least, and CHH the most common sequence motifs, this is consistent with a parsimonious explanation that the overall density of CG sites predict the overall correlation with non-conversion rates. In summary, it appears that non-conversion errors vary between regions of the genome in a systematic way. Although the underlying reasons for this are unclear, this observation warrants caution in using a single point estimate to account for errors across the genome.

### Errors are consistent with Tn5 overactivity

Summarising results so far, our observations indicate that non-conversion errors occur at high frequency in samples from diverse organisms, but that error rates vary between reads and across the genome. It appears that affected reads result from two distinct processes that generate completely and partially unconverted reads, with a bias towards non-conversion of cytosines towards the 3' ends of reads. These observations are consistent with the hypothesis that the Tn5 transposase creates single-strand nicks, which are repaired by strand displacement by the *Bst* polymerase in the presence of 5-methyl-dCTPs. Lu et al. [7] hypothesised that Tn5 nicks the adaptor sequence, leading to displacement of the entire fragment. This model

would account for those reads where every cytosine is unconverted (Fig 1B). We speculate that the partially non-converted reads are due a second nicking process that occurs randomly throughout the body of a DNA fragment. Because the *Bst* polymerase replicates in 5' to 3' direction, random nick positions would systematically bias non-conversion towards downstream cytosines. This is consistent with the observation that non-conversion errors are more common towards the 3' end of reads (Fig 1C). Tn5 is known to preferentially insert at guanine and cytosine positions [11,12], which is consistent with the correlation between non-conversion rates and GC content (Fig 2C). Moreover, there is evidence that Tn5 insertion sites tend to be spaced at 5-bp intervals [11], which matches the periodicity in non-conversion errors in *D. melanogaster* (Fig 1C). However, it is not known whether patterns of insertion site bias reflect the same process as a bias in the location of single-strand nicks. Although these data do not allow us to say with certainty what is causing non-conversion errors, this model of the combined action of single-strand nicks followed by strand displacement seems to be a reasonable working hypothesis to explain non-conversion errors.

It is worth considering the implications of this model for other applications. The fact that strand-displacement protocols use methylated dCTPs but oligo-replacement protocols use unmethylated dCTPs provides a straightforward explanation for why the former have substantial conversion errors while the latter apparently do not. However, both approaches involve Tn5 transposase, and it stands to reason that single-strand breaks should occur with similar frequency in both. If these are repaired with unmethylated dCTPs this would introduce a bias in the opposite direction to that described here that would *reduce* apparent methylation. Importantly, such a bias would almost certainly go undetected unless a completely methylated control sequence were included. If our working hypothesis is correct, it is unlikely that any such bias would be of the same magnitude as the non-conversion errors we describe because oligo-replacement protocols do not involve strand displacement. Nevertheless we suggest that some caution is warranted regarding possible false-conversion errors in standard tagmentation protocols.

How can we deal with non-conversion errors in the light of this working hypothesis? Suzuki et al. [9] filtered reads if they have multiple unconverted CHH sites (H is any nucleotide except G). This makes sense in mammals where methylation is primarily on CG sites, but will not work for plants where CHH methylation is both real and interesting. Lu et al. [7] scored reads based on the number and position of unconverted cytosines, and discard suspicious reads. However, the logic of this score is not clear, and was developed with completely non-converted reads in mind. It will likely not work in the presence of partially unconverted reads. An alternative partial fix would be to filter out completely unconverted reads, but this would not address the partially unconverted reads, and risks biasing inferred methylation in regions that are truly highly methylated. Fortunately, since non-conversion errors happen in vitro and not in vivo we can account for them by modelling the error-generating process statistically. In the sections that follow and the Materials and Methods we describe a framework to model non-conversion errors and apply it to experimental data from *A. thaliana*.

## Methylation estimates depend more on sample size than non-conversion errors

Often one may wish to estimate and compare average methylation within and between regions of the genome. Estimates of average methylation reflect a binomial sampling process of converted and unconverted cytosines from different cells. We aim to estimate this mean with as little bias and sampling variance as possible. Non-conversion of truly unmethylated

cytosines will bias apparent methylation upwards. Likewise, incorrect conversion of truly methylated cytosines will bias methylation downwards. In this study we focus on the former prcoess, but note that the latter is distinctly plausible. Since these errors are statistical rather than biological, we can account for them by allowing for the probabilities of errors in the binomial sampling process (see Methods). Given a good estimate of error rates, it is straightforward adjust estimates to correct these biases.

However, this correction is complicated by two practical concerns that effectively increase sample variance. First, as noted above, there seems to be substantial variation in error rates between windows of the chloroplast. If we try to use the chloroplast to gauge error rates at the rest of the genome, this implies that there is substantial uncertainty around the error rate at any specific window. This will tend to overestimate true methylation when the error rate is above the mean error rate, and vice versa. Second, the binomial sampling process itself introduces uncertainty about true methylation level. This is especially true when the number of 'binomial trials' is small, such as when only a small region of the genome is examined and/or when sequencing coverage is low. Variation in error rates and sample size should be expected to hinder our ability to estimate mean methylation accurately.

We used simulations to investigate the practical significance of conversion errors and binomial sampling on estimates of mean methylation. We simulated converted and unconverted reads on transposable elements (TEs) of typical size with and without non-conversion errors, and attempt to recover true methylation levels. We also summed reads across multiple loci, each with a unique error rate, to reflect summing over multiple TEs in the same region or family (e.g. [13]). When data contain no non-conversion errors (red points in Fig 3A and 3B) any deviation from the true mean reflects variance due to binomial sampling only, and represents the best-case scenario for given coverage and number of loci. When non-conversion errors are introduced but the error rate is known we observe an increase in variation around the true mean (green). This additional error must reflect the additional binomial variance that arises from sampling from both truly methylated and unmethylated reads. When the error rate is not known but is estimated (blue) estimated means are even less accurate. This demonstrates that both binomial sampling and uncertainty in the true error rate contribute to uncertainty in true methylation rate.

Despite this, the effect of error rates on accuracy was much smaller than the effect of sample size. The accuracy of estimated means increased dramatically with increased number of trials, either by increasing the number of loci or sequence coverage, including when data are simulated without errors (Fig 3A and 3B). This is not surprising in itself, but it is notable that a fairly simple correction for error rates can recover good estimates of mean methylation over thousands of cytosines using modest sequence coverage, even with relatively high non-conversion errors. This implies that the most practical way to ensure accurate estimates of methylation is to increase sample size, either by increasing sequencing depth or by summing over larger windows or multiple loci. Since sequencing depth is typically limited by budgets, the latter option is likely the most practical, and often makes biological sense. For example, multiple TEs may be targeted and co-regulated by the same silencing pathways, so it may be logical to examine them together. Increasing the number of binomial trials is the surest path to accurate estimates of mean methylation, even when errors are common.

Aside from any effects on the variance, cytosine non-conversion errors should systematically increase the apparent number of methylated cytosines, and bias estimates of methylation level upwards if ignored. However, our estimates of mean methylation level after adjustment for non-conversion errors are symmetrical around the true value (Fig 3A) and show no evidence for systematic bias (Fig 3C). Moreover, the scale of any bias is an order magnitude less than that of overall deviations (Fig 3B and 3C). These observations are true even when there

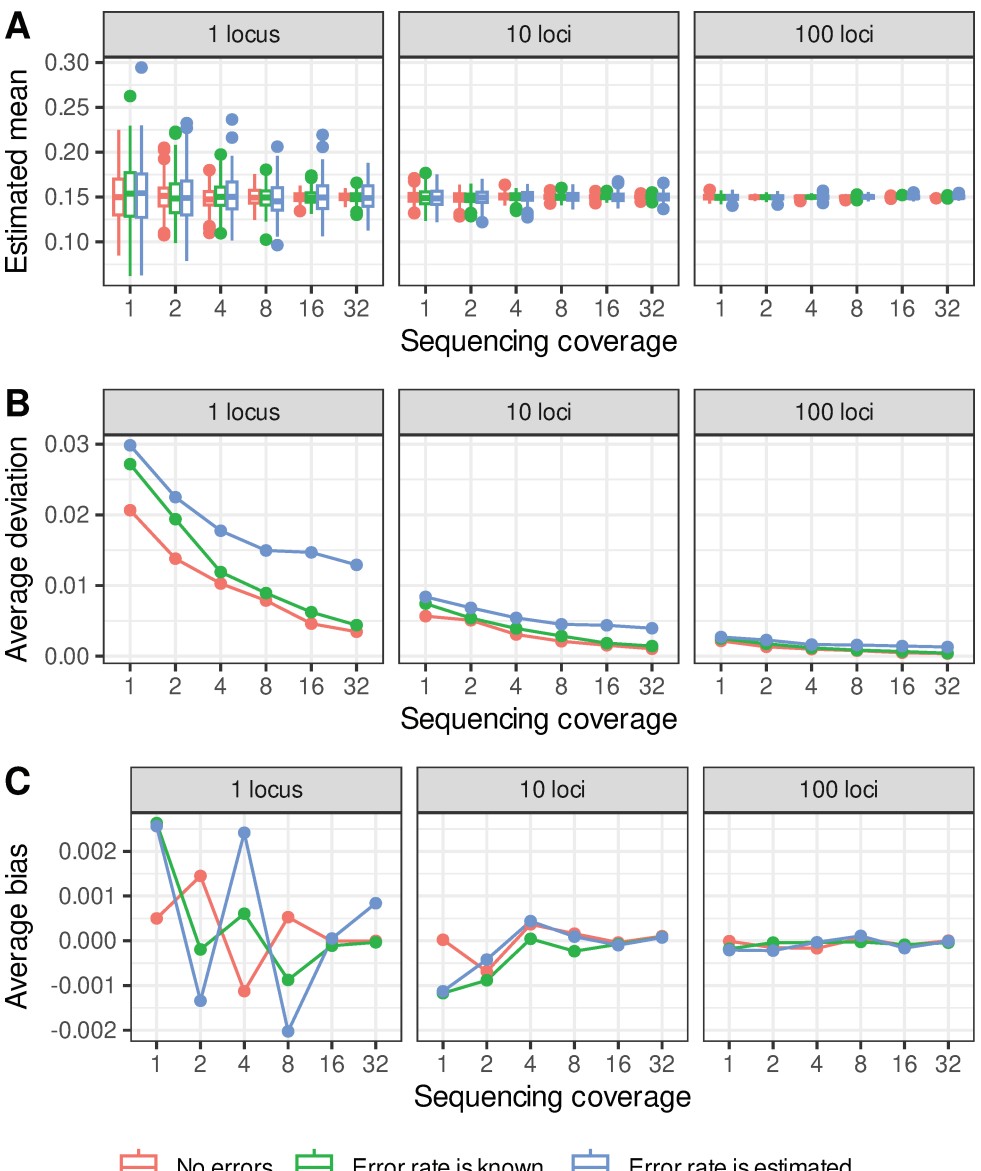

**Fig 3. Estimates of mean methylation in simulated data using different estimates of the non-conversion error rate.** Plots show the effect of combining data over 1, 10 or 100 loci with 200 cytosines each for increasing sequencing depth. True methylation is 0.15 for all loci. Error rates for each locus are either zero or are drawn from a beta distribution with shape parameters $a = 17$ and $b = 220$. Methylation level is estimated for data with no errors, for data with errors using the true error rate for the simulation, and for data with errors when the error rate is estimated as the mean of the beta distribution. (**A**) Distribution of estimated means across replicate 200 simulations. (**B**) Mean deviation from the true value across simulations. Note that points are offset along the x-axis for visual clarity. (**C**) Mean bias across simulations, calculated as the observed estimated methylation level minus true methylation level.

is uncertainty about the true error rate. This indicates that the statistical correction for non-conversion errors is effective in removing the bias, and that sampling variance in estimated methylation level is a greater concern than bias caused by non-conversion errors.

## Imputation of methylation state

In some applications one might wish to classify a discrete methylation 'state', rather than quantify methylation level. For example, if the biological question addresses whether a locus is targeted by silencing machinery or not it may be practical to classify a region as 'methylated' or 'unmethylated', rather than try to deal with noisy or overdispersed quantitative estimates. To illustrate this we categorised methylation status of regions of the *A. thaliana* genome based on a model of methylation homeostasis posited by Zhang et al. [14]. In plants, cytosines can be methlylated in all three sequence contexts (CG, CHG and CHH) maintained by distinct pathways [15]. Although much of the *A. thaliana* genome is unmethylated, some genes display 'gene-body methylation' characterised by methylation of CG sites, whose origin and function has attracted much attention [16]. Transposable elements are associated with methylation in all three of these contexts [3,17]. Zhang et al. [14] describe a continuum from unmethylated to gene-body-like methylation (GBM-like) to TE-like methylation. Since these three states are likely maintained by distinct biological pathways it makes sense to model them as distinct methylation states.

We categorised methylation status for bisulphite data in the presence of substantial non-conversion errors in our *A. thaliana* samples. We used a model that compares the evidence that the counts of observed converted and unconverted reads in the three sequence contexts are consistent with non-conversion errors only, or a mixture of errors and real cytosine methylation, corresponding to truly zero and non-zero methylation levels respectively. We applied this model to (1) windows of the chloroplast, which should be unmethylated; (2) annotated transposable elements known to be actively regulated by the CMT2 and RdDM pathways [18] and likely to be TE-like methylated; (3) coding genes, likely reflecting a mix of all three methylation states [14]. For each window/locus we quantified the evidence that DNA was unmethylated, GBM-like methylated, or TE-like methylated.

The observed states of the chloroplast and TEs showed a strong tendency towards their expected state (Fig 4). 95.6% of windows on the chloroplast were most likely to be unmethylated. Likewise, 94.9% of CMT2-targetted TEs and 86.9% of RdDM-targetted TEs were TE-like methylated. We found nearly identical results when we sum over the probabilities that each window/TE is in each state rather than make hard calls about methylation state (Fig 4). The methylation state of unmethylated DNA and transposable elements can be inferred despite non-conversion errors.

Methylation states of genes were also qualitatively similar to previous reports. 47.8% of genes were most likely to be unmethylated, 36.9% GBM-like methylated and 15.3% TE-like methylated. This matches the general pattern of methylation states reported by Zhang et al. [14], although that study inferred approximately fifty percent more unmethylated genes and around half as many GBM-like and TE-like genes. In contrast to the chloroplast and TEs, it is not clear what the expectation should be, and gene methylation states likely reflect a dynamic equilibrium [14]. Non-conversion errors in these data almost certainly contribute this quantitative discrepancy, but they are also likely driven to a large part by differences in how methylation states are called. States inferred by Zhang et al. [14] were based on an approach described by Takuno and Gaut [19] that compares methylation at each gene to the genome-wide average via a binomial test. This test is conservative, in that only the most strongly gene-body methylated genes will show a significant increase in methylation, which will tend to increase the apparent incidence of unmethylated genes. Moreover, it is not clear that comparing methylation at individual loci to a genome-wide average is an appropriate null hypothesis. For example, if there were a genetic or environmental modifier that increased global methylation it would in fact be *more* difficult to identify individual loci with strong

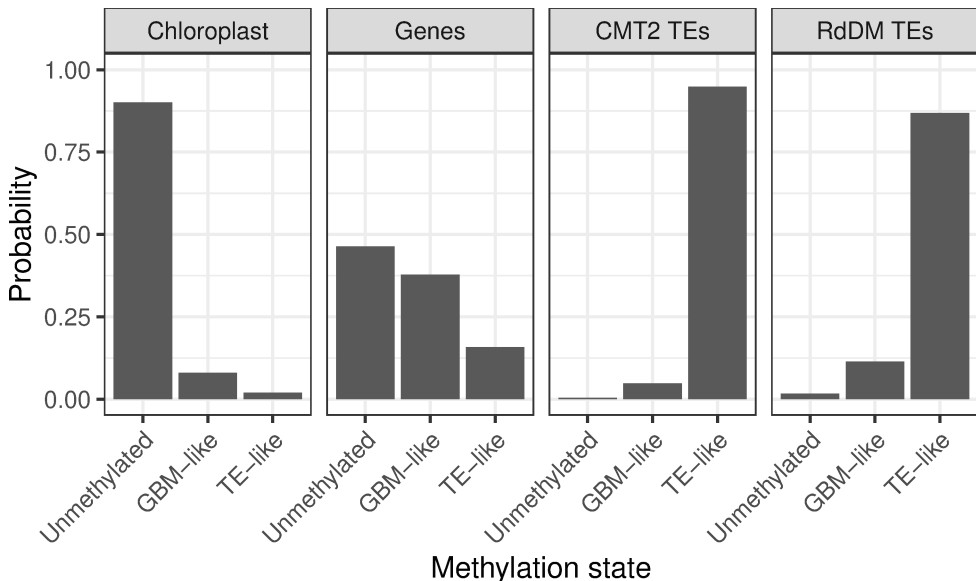

**Fig 4. Methylation state calls in windows of the chloroplast, genes, CMT-targetted TEs and RdDM-targetted TEs.**
Plots show the posterior probabilities that a single window/gene/TE is in each state. Only those features with at least one aligned read in each sequence context are shown.

methylation. In contrast, our approach quantifies the evidence for competing hypotheses (possible methylation states) based on the evidence available for each locus on its own [20,21].

## Conclusions

Strand-displacement-based tagmentation protocols allow for bisulphite sequencing of samples with smaller amounts of starting material, and are more easily scalable to high-throughput samples than previous protocols. However, this scalability comes at the cost of substantial cytosine non-conversion errors. The patterns of these errors is consistent the Tn5 transposase making single-strand nicks across fragments, followed by strand displacement of the rest of the downstream fragment by *Bst* polymerase, replacing every cytosine with 5-methyl-dCTPs. Fortunately, these errors can be modelled and corrected statistically. We demonstrate that non-conversion errors have less of an effect on quantitative estimates of methylation than does binomial sampling, which is common to all protocols. Ultimately the additional work in dealing with errors should be traded off against the ease of scalability of strand-displacement-based bisulphite protocols.

## Materials and methods

### Bisulphite sequencing

We performed bisulphite sequencing of DNA from *A. thaliana*, *D. melanogaster* and phage $\lambda$. We extracted DNA from the *A. thaliana* accession Col-0 using the NucleoMag 96 Plant kit (Macherey Nagel Bioanalysis REF. 744400.24). An aliquot of *D. melanogaster* DNA extracted from ovarian somatic cell culture [22,23] was kindly donated by Dominik Handler. We prepared bisulphite libraries based on the protocol described by Weichenhan et al. [8] with some modifications. We used 20-30 ng of genomic DNA per sample. As an internal control, we added 5 pg of unmethylated DNA from phage $\lambda$ (New England Biolabs, E7123A) per 10 ng

sample DNA. We performed tagmentation using in-house Tn5 transposase [24] in ratios of 1:1 and 1:2 transposase to DNA. We used EZ-96 DNA Methylation-Gold MagPrep (Zymo, D5043) for bisulphite conversion. We did 13 or 15 cycles of PCR amplification for enrichment, and we used Illumina Nextera DNA indexes (i7/i5) to allow multiplexing. We performed size selection and PCR clean-up with in-house SPRI beads. We validated libraries by Fragment Analyzer™ Automated CE System (Advanced Analytical) and pooled them in equimolar concentration. Median insert size was around 350 bp.

Libraries were sequenced by the Next Generation Sequencing Facility at Vienna BioCenter Core Facilities (VBCF), on an Illumina NovaSeq Analyzer using manufacturer's standard cluster generation ad sequencing tools (S2 flowcell with 100 bp paired end reads). To compensate for low nucleotide diversity in bisulphite libraries, each lane included a 15% splike in of phiX Control.

## Bioinformatics

We used cutadapt to remove adaptor sequences from raw reads [25]. We also trimmed 15 bp from the 5' and 9 bp from the 3' end of each read corresponding to the gap-repair regions of each fragment and visually inspected the quality of unaligned reads using FastQC [26]. We used Bismark to align reads to the TAIR10 and *D. melanogaster* 6 reference genomes [27]. We concatenated both reference genomes with the phage $\lambda$ genome. In addition, we included the genomes of the microbial symbionts *Acetobacter aceti* and *Lactobacillus acidophilus*. We used Bismark to remove PCR-duplicated sequences and calculate the number of converted and unconverted cytosines aligning to each genomic cytosine position, and used the resulting cytosine report files to quantify overall non-conversion rates across the Drosophila and phage $\lambda$ genomes.

## Data analysis

We used custom Python functions to quantify patterns of non-conversion within reads and across the genome, available via the package *methlab* [28]. For *A. thaliana* we estimated non-conversion on the first half of the chloroplast only, because the second half shows strong homology with one of the autosomes.

We carried out statistical analyses in *R* 4.3.3 using *RStudio* 2023.06.1 [29,30] and *ggplot2* for plotting [31]. To investigate patterns of non-conversion rates between windows of the *A. thaliana* chloroplast we modelled (logit) non-conversion rate with the numbers of CG, CHG and CHH sites as fixed effects, and a random effect of library ID as a random effect using *lme4* [32]. The logit transformation accounts for the non-linearity of variance that occurs when data are bounded by zero and one. We used the R package *r2glmm* 0.1.2 to estimate the variance explained by each fixed effect [33].

The computational results presented were obtained using the CLIP cluster (https://clip.science).

## Statistical models to account for conversion errors

**Error rates in a binomial model** Given an alignment of bisulphite-converted reads to a genome we observe a total of $n$ reads mapping to genomics cytosines within a given region. Of these, $y$ reads are observed to be unconverted cytosines and $n–y$ reads appear as converted thymines. The goal is to estimate the true mean methylation level $\theta$ which generated these data, accounting for conversion errors.

In the absence of errors, the likelihood of the data given $\theta$ is binomially distributed as

$$\Pr(y|\theta) = \binom{n}{y} \theta^y (1-\theta)^{n-y} \tag{1}$$

with mean $\theta = y/n$. Data are not perfect, so we would like to incorporate two error terms:

- $\epsilon_1$ is the probability that a truly unmethylated cytosine appears methylated (the bisulphite non-conversion rate).
- $\epsilon_2$ is the probability that a truly methylated cytosine appears unmethylated, which we include for completeness.

Reads observed as cytosines may thus be either truly methylated with probability $\theta(1-\epsilon_2)$ or truly unmethylated with probability $(1-\theta)\epsilon_1$. Likewise, reads observed as thymine may be either truly unmethylated with probability $(1-\theta)(1-\epsilon_1)$ or truly methylated with probability $\theta\epsilon_2$. This changes the likelihood to

$$\Pr(y|\theta, \epsilon_1, \epsilon_2) = \binom{n}{y} [\theta(1-\epsilon_2) + (1-\theta)\epsilon_1]^y$$
$$[\theta\epsilon_2 + (1-\theta)(1-\epsilon_1)]^{n-y} \tag{2}$$

Summarising $p = [\theta(1-\epsilon_2) + (1-\theta)\epsilon_1] = y/n$ for brevity, this has a closed-form maximum-likelihood estimate

$$\hat{\theta} = \frac{\epsilon_1 - p}{\epsilon_1 + \epsilon_2 - 1} \tag{3}$$

Note that $\hat{\theta} < 0$ when $\epsilon_1 > \theta$ and $\hat{\theta} > 1$ when $\epsilon_2 > (1-\theta)$, which are not valid results. In these cases we set $\hat{\theta}$ to zero and one respectively.

**Accounting for uncertainty in error rate** The above formulation is valid for the case where there is a clear point estimate for error rates. If there is substantial uncertainty around these estimates, and especially if they are likely to vary across the genomes, then error rates can be modelled as coming from beta distributions with shape parameters $a$ and $b$. For DNA which is known to be unmethylated, such that all unconverted cytosines are incorrect:

$$\Pr(p|a, b) = \frac{p^{a-1}(1-p)^{b-1}}{B(a, b)} \tag{4}$$

where $B(a, b)$ is the Beta function. The shape parameters can be estimated by method-of-moments using estimates of the sample mean $\mu$ and variance $\sigma$ of the distribution. For example, one can calculate mean ($\mu$) and variance ($\sigma^2$) in the proportion of unconverted cytosines across windows of DNA known to be unmethylated. Estimates for shape parameters are then:

$$\hat{a} = \mu\left(\frac{\mu(1-\mu)}{\sigma^2} - 1\right) \tag{5}$$

$$\hat{b} = (1-\mu)\left(\frac{\mu(1-\mu)}{\sigma^2} - 1\right) \tag{6}$$

**Classification of methylation state** In some cases it makes biological sense to classify the 'state' of a region as methylated or unmethylated rather than quantify methylation level. If we assume false conversion errors ($\epsilon_2$) are negligible, this can be achieved by comparing the

evidence that the proportion of unconverted cytosines $p$ is drawn from distribution $f(p)$ generating false non-conversion errors, or from an alternative distribution $g(p)$ that reflects real biological processes.

First, $f(p)$ can be described as a combination of the binomial and beta processes described above (Eqs 2 and 4). If all the reads are truly unmethylated then $\theta = 0$, and hence

$$f(p) = \Pr(y|p, n)\Pr(p|a, b) \tag{7}$$

where $a, b$ are shape parameters of the beta distribution of error rates. This accounts for both the binomial process of drawing a finite sample of reads, and the uncertainty in error rate.

It may be possible to modify Eq 7 to completely describe the distribution of truly methylated cytosines. In this case $\epsilon_1 = \epsilon_2 = 0$, but $\theta > 0$, and hence

$$g(p) = \Pr(y|p, n)\Pr(p|a^*, b^*) \tag{8}$$

where $a^*$ and $b^*$ are the beta shape parameters for the distribution of true methylation levels. This is only valid if it is possible to estimate these shape parameters, for example from external data from libraries prepared using a different protocol for which conversion error rates are negligible.

However, as long as the true methylation level is substantially larger than error rates, we can simplify the problem by asking whether observed unconverted reads are compatible with non-conversion errors alone, or with more methylation than that. A simpler approach is to model

$$g(p) = \Pr(y|p, n)\Pr(p \leq P|a, b) \tag{9}$$

where $\Pr(p \leq P|a, b)$ is the cumulative density function of the beta distribution of error rates. This approaches zero when the proportion of unconverted reads is small, and approaches one when the number of unconverted reads is more than can be accounted for by $f(p)$, and thus readily distinguishes 'high' and 'low' non-conversion rates. This has the advantage that we need only know $a$ and $b$, which can be estimated in the presence of cytosine non-conversion errors. For this reason, we use this estimate of $g(p)$ in this manuscript.

We can apply this model to quantify the evidence that a region of DNA is unmethylated (um), gene-body-like methylated (gbm) or TE-like methylated (TEm) based on the observed proportions of non-converted reads in the CG, CHG and CHH sequence contexts ($p_{CG}$, $p_{CHG}$ and $p_{CHH}$; H is any nucleotide apart from G). In unmethylated DNA all unconverted cytosines are due to non-conversion errors, and hence the likelihood that the region is unmethylated is

$$L_{um} = f(p_{CG})f(p_{CHG})f(p_{CHH}) \tag{10}$$

In gene-body-like-methylated DNA, CG sites are methylated but CHG and CHH sites are not:

$$L_{gbm} = g(p_{CG})f(p_{CHG})f(p_{CHH}) \tag{11}$$

In TE-like-methylated DNA cytosines are methylated in all contexts:

$$L_{TEm} = g(p_{CG})g(p_{CHG})g(p_{CHH}) \tag{12}$$

These likelihoods can be converted to probabilities by dividing by the sum of the three likelihoods. The global probability that any single locus is in methylation state $i$, accounting

for the uncertainty in state at each locus, is simply the average probability of being in state $i$ over all loci.

## Simulations

We used simulations to investigate how well we can estimate true mean methylation when observed data are contaminated with non-conversion errors. We simulated loci of 800 nucleotides (200 cytosines), corresponding to the average size (798bp) of a transposable element or TE fragment in the *A. thaliana* TAIR10 annotation, with 1, 2, 4, 8, 16 or 32 reads mapping to each cytosine. We simulated the conversion state of each read at each cytosine as a binomial process with the total number of trials being the number of cytosines multiplied by coverage per cytosine. Following Eq 2 and ignoring false conversion errors, each read may be observed as unconverted with probability $p = \theta + (1 - \theta)\epsilon_1$, where $\theta$ is the probability that the read is truly methylated and $\epsilon_1$ is a non-conversion error rate. We used $\theta = 0.15$ for these simulations, which is both a realistic estimate of TE methylation level [34]. We note that much higher methylation levels would be expected for CpG islands or gene-bodies; we use the lower value because this represents the more challenging case where real methylation levels are close to non-conversion error rates.

For each locus we drew a value for $\epsilon_1$ from a beta distribution describing variation in error rates across the genome. To estimate biologically realistic shape parameters for this distribution we estimated the mean and variation in non-conversion rates in 150 bp windows across the chloroplast for one of the *A. thaliana* samples, and calculated beta shape parameters by method of moments (Eqs 5 and 6).

## Acknowledgements

We thank Robert Schmitz, Dieter Weichenhan and members of the Nordborg group for discussion of the results. We specifically thank Yoav Voichek for technical assistance on how to disembowel SAM files. We also thank Dominik Handler for donated an aliquot of *D. melanogaster* DNA.

## Author contributions

**Conceptualization:** Thomas James Ellis, Viktoria Nizhynska, Almudena Mollá-Morales.

**Data curation:** Thomas James Ellis, Rahul Pisupati.

**Formal analysis:** Thomas James Ellis, Rahul Pisupati.

**Funding acquisition:** Magnus Nordborg.

**Investigation:** Thomas James Ellis.

**Methodology:** Viktoria Nizhynska.

**Project administration:** Almudena Mollá-Morales.

**Resources:** Almudena Mollá-Morales.

**Software:** Thomas James Ellis.

**Supervision:** Magnus Nordborg.

**Visualization:** Thomas James Ellis.

**Writing – original draft:** Thomas James Ellis.

**Writing – review & editing:** Viktoria Nizhynska, Rahul Pisupati, Almudena Mollá-Morales, Magnus Nordborg.

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
