## [Decision Letter · Decision Letter 0]

23 Jan 2025

PONE-D-24-42340Bisulphite sequencing in the presence of cytosine-conversion errorsPLOS ONE

Dear Dr. Ellis,

Thank you for submitting your manuscript to PLOS ONE. After careful consideration, we feel that it has merit but does not fully meet PLOS ONE’s publication criteria as it currently stands. Therefore, we invite you to submit a revised version of the manuscript that addresses the points raised during the review process.

We look forward to receiving your revised manuscript.

Kind regards,

Osman El-Maarri, Ph.D

Academic Editor

PLOS ONE

Journal Requirements:

“This work was supported by an ERC Advanced Grant (789037) to Magnus Nordborg.”

Reviewers' comments:

Reviewer's Responses to Questions

**Comments to the Author**

1. Is the manuscript technically sound, and do the data support the conclusions?

Reviewer #1: Partly

2. Has the statistical analysis been performed appropriately and rigorously? 

Reviewer #1: No

3. Have the authors made all data underlying the findings in their manuscript fully available?

Reviewer #1: Yes

4. Is the manuscript presented in an intelligible fashion and written in standard English?

Reviewer #1: No

5. Review Comments to the Author

Reviewer #1: The study aims to address the errors and bias in bisulfite cytosine conversion, the most exclusively used method of DNA methylation analysis. Unmethylated DNA (chloroplast DNA, D. melanogaster DNA, and Lambda DNA) was subjected to Tn5 tagmentation and bisulfite sequencing. The non-converted cytosine rate is shown to be high and varies depending on DNA issues (Fig. 1) and DNA sequences (Fig. 2). The reviewer has several comments and observations.

1) The non-conversion rate shown in Fig. 1A differed among tested DNAs, which was then explained by technical variations between libraries rather than biological variation. Could this variation have been influenced by the DNA input amount and the circular/linear form of DNA for bisulfite reactions or by the GC ratio and CG motifs (CCGG, GGCC, CGCG) on the DNA itself? The authors used 20-30 ng of DNA (genomic Arabidopsis DNA, genomic D. melanogaster DNA and Lambda DNA) with different sizes corresponding to different copy numbers that were subsequently increased by PCR amplification before bisulfite conversion. However, it has been reported that conversion efficiency is affected by the copy number of the target sequence, not the DNA amount. As seen in Fig. 1A-B, non-conversion errors are lowest for Arabidopsis DNA and highest for Lambda DNA (~50 kb).

2) A bias of non-converted cytosines towards the 3’ ends of reads was explained by using the Bst polymerase with methylated dCTP to fill up random single-strand nicks made by Tn5 transposase in the 5’ to 3’ direction. However, this bias was too strong (Fig. 1C). It has been reported that the sequence complexity, the GC content, secondary structure elements and even a certain cytosine in a particular sequence can interfere with bisulfite conversion efficiency. Could this bias have been due to being flanked by fully and densely methylated adaptor sequences whose pattern may not be present in the genome?

3) The authors suggested that two distinct processes might have contributed to incorrect cytosine non-conversions. In actuality, two processes are the reasonably typical results from technical errors in bisulfite conversions: the inefficient denaturation caused by an excessive input amount of DNA and the partial conversion commonly due to complex DNA structures. Most reads (71% on average) showed no unconverted cytosines, likely due to the inefficient denaturation during bisulfite processing causing a downward bias in methylation levels instead of an upward bias observed in partial conversion. Moreover, an unusually high number of unconverted unmethylated cytosines was explained by single-strand nicks by the transposase. As such, the non-converted cytosine errors were likely technical rather than statistical, making them not reliably adjusted by a statistical framework to quantify and classify actual cytosine methylation.

4) In Fig. 2, the high-fold variation in cytosine non-conversion rate between 150bp-windows should be further clarified by comparisons between a narrower number of windows to at least exemplify a contributor to this variation, such as CG motifs (CCGG, GGCC, CGCG) or short stretches of non-CpG cytosines.

In Eq. 2 with the probability p = θ + (1 −θ) ϵ 1, where θ = y/n, the authors failed to address how θ= 0.15, chosen based on a realistic estimate of TE methylation level reported in Ref. 26, is applicable when assessing the methylation levels of CpG islands and gene bodies.

There is no Fig. 2D.

6. PLOS authors have the option to publish the peer review history of their article (what does this mean?). If published, this will include your full peer review and any attached files.

Reviewer #1: No

---

## [Author Response · Author response to Decision Letter 1]

26 Feb 2025

Reviewer #1: The study aims to address the errors and bias in bisulfite cytosine conversion, the most exclusively used method of DNA methylation analysis. Unmethylated DNA (chloroplast DNA, D. melanogaster DNA, and Lambda DNA) was subjected to Tn5 tagmentation and bisulfite sequencing. The non-converted cytosine rate is shown to be high and varies depending on DNA issues (Fig. 1) and DNA sequences (Fig. 2). The reviewer has several comments and observations.

1) The non-conversion rate shown in Fig. 1A differed among tested DNAs, which was then explained by technical variations between libraries rather than biological variation. Could this variation have been influenced by the DNA input amount and the circular/linear form of DNA for bisulfite reactions or by the GC ratio and CG motifs (CCGG, GGCC, CGCG) on the DNA itself? The authors used 20-30 ng of DNA (genomic Arabidopsis DNA, genomic D. melanogaster DNA and Lambda DNA) with different sizes corresponding to different copy numbers that were subsequently increased by PCR amplification before bisulfite conversion. However, it has been reported that conversion efficiency is affected by the copy number of the target sequence, not the DNA amount. As seen in Fig. 1A-B, non-conversion errors are lowest for Arabidopsis DNA and highest for Lambda DNA (~50 kb).

We have rearranged the order of the text to clarify that we used 20-30ng of genomic DNA for A. thaliana and D.melanogaster, and added 5pg phage lambda DNA per 10ng genomic DNA to each library (lines 325-326). This is typical of bisulphite protocols, and usually does not result in high non-conversion rates in ‘standard’ protocols that do not use strand displacement and methylated dCTPs.

Furthermore, although sample sizes preclude any meaningful statistical test, we did not find any apparent effect of the amount of input DNA. Rather, we note that the Drosophila and lambda samples with the highest non-conversion rates were from the same libraries, which strongly indicates that the major differences are between libraries. We have added a sentence highlighting this point (line 66-69).

2) A bias of non-converted cytosines towards the 3’ ends of reads was explained by using the Bst polymerase with methylated dCTP to fill up random single-strand nicks made by Tn5 transposase in the 5’ to 3’ direction. However, this bias was too strong (Fig. 1C). It has been reported that the sequence complexity, the GC content, secondary structure elements and even a certain cytosine in a particular sequence can interfere with bisulfite conversion efficiency. Could this bias have been due to being flanked by fully and densely methylated adaptor sequences whose pattern may not be present in the genome?

This is also unlikely, because the methylated adaptor sequences occur at both ends of the read, so we would not expect to see an increase in non-conversion from one end to the other end of the read. We have clarified this point in the text (lines 87-88).

3) The authors suggested that two distinct processes might have contributed to incorrect cytosine non-conversions. In actuality, two processes are the reasonably typical results from technical errors in bisulfite conversions: the inefficient denaturation caused by an excessive input amount of DNA and the partial conversion commonly due to complex DNA structures.

In previous studies we have used diverse lines of A. thaliana that are likely to have much more variation in structural complexity than used in this paper (eg Kawakatsu et al., 2016, Cell 166:492-505) and using similar concentrations of DNA. Crucially, these studies used bisulphite protocols that did not involve strand displacement or methylated dCTPs, and apparent cytosine non-conversion rates estimated in the same way were <1%. As such, these processes are unlikely to explain our observations.

Most reads (71% on average) showed no unconverted cytosines, likely due to the inefficient denaturation during bisulfite processing causing a downward bias in methylation levels instead of an upward bias observed in partial conversion.

We refer only to reads mapping to Arabidopsis chloroplast, the D. melanogaster genome and lambda phage, which are expected to have zero truly methylated sites (lines 54-56).

Moreover, an unusually high number of unconverted unmethylated cytosines was explained by single-strand nicks by the transposase. As such, the non-converted cytosine errors were likely technical rather than statistical, making them not reliably adjusted by a statistical framework to quantify and classify actual cytosine methylation.

Statistical distributions describe the stochastic processes arising in nature. For example, when we estimate the proportion of methylated or unmethylated cytosines at a single site we are taking an average over a finite sample of cells, so there is some sampling variance that must occur from sample to sample. The binomial distribution describes this sampling process well, and has been widely used to model methylation levels from bisulphite data. We agree that cytosine non-conversion arises from technical errors, but these errors simply alter the probability that a cytosine on a read observed to be methylated is truly methylated or not. The section beginning on line 365 models exactly that adjustment, and the simulation results validate that these models work as expected. As such, technical errors can be well described by common statistical distributions.

4) In Fig. 2, the high-fold variation in cytosine non-conversion rate between 150bp-windows should be further clarified by comparisons between a narrower number of windows to at least exemplify a contributor to this variation, such as CG motifs (CCGG, GGCC, CGCG) or short stretches of non-CpG cytosines.

We now clarify that the reason for using 150bp windows is that this corresponds to the approximate number of base pairs in a nucleosome loop (lines 103-105). We have repeated the analysis with 11bp windows, corresponding to the length of one helix turn, and find very similar results (line 115-116). In addition, we quantified the proportion of variation in non-conversion rates between windows that is explained by the densities of CG, CHG and CHH sites, and find that the variation is likely due simply to the overall density of cytosines (lines 125-129). We have also highlighted that this correlation may be related to Tn5’s insertion preference for GC sites, noting that insertion is not the same thing as single-stranded nicks (149-155).

In Eq. 2 with the probability p = θ + (1 −θ) ϵ 1, where θ = y/n, the authors failed to address how θ= 0.15, chosen based on a realistic estimate of TE methylation level reported in Ref. 26, is applicable when assessing the methylation levels of CpG islands and gene bodies.

We have clarified that we use the lower value because this represents the more challenging case where error rates are very close to what we would expect real methylation levels to be (lines 452-456). If one were looking at regions with much higher true methylation levels, non-conversion errors make a much smaller contribution to the estimated mean, and are much easier to distinguish from errors.

There is no Fig. 2D.

We have corrected this cross-reference to figure 2C.

---

## [Decision Letter · Decision Letter 1]

24 Mar 2025

Bisulphite sequencing in the presence of cytosine-conversion errors

PONE-D-24-42340R1

Dear Dr. Ellis,

We’re pleased to inform you that your manuscript has been judged scientifically suitable for publication and will be formally accepted for publication once it meets all outstanding technical requirements.

Kind regards,

Osman El-Maarri, Ph.D

Academic Editor

PLOS ONE

Additional Editor Comments (optional):

Reviewers' comments:

Reviewer's Responses to Questions

**Comments to the Author**

1. If the authors have adequately addressed your comments raised in a previous round of review and you feel that this manuscript is now acceptable for publication, you may indicate that here to bypass the “Comments to the Author” section, enter your conflict of interest statement in the “Confidential to Editor” section, and submit your "Accept" recommendation.

Reviewer #1: (No Response)

2. Is the manuscript technically sound, and do the data support the conclusions?

Reviewer #1: (No Response)

3. Has the statistical analysis been performed appropriately and rigorously? 

Reviewer #1: (No Response)

4. Have the authors made all data underlying the findings in their manuscript fully available?

Reviewer #1: (No Response)

5. Is the manuscript presented in an intelligible fashion and written in standard English?

Reviewer #1: (No Response)

6. Review Comments to the Author

Reviewer #1: (No Response)

7. PLOS authors have the option to publish the peer review history of their article (what does this mean?). If published, this will include your full peer review and any attached files.

Reviewer #1: No

---

## [Editor Report · Acceptance letter]

PONE-D-24-42340R1

PLOS ONE

Dear Dr. Ellis,

I'm pleased to inform you that your manuscript has been deemed suitable for publication in PLOS ONE. Congratulations! Your manuscript is now being handed over to our production team.

Kind regards,

on behalf of

Priv.-Doz. Dr. Osman El-Maarri

Academic Editor

PLOS ONE